# Incipient ferroelectricity of water molecules confined to nano-channels of beryl

B.P. Gorshunov[1,2,3], V.I. Torgashev[4], E.S. Zhukova[1,2,3], V.G. Thomas[5], M.A. Belyanchikov[1], C. Kadlec[6], F. Kadlec[6], M. Savinov[6], T. Ostapchuk[6], J. Petzelt[6], J. Prokleška[7], P.V. Tomas[8,9], E.V. Pestrjakov[10], D.A. Fursenko[5], G.S. Shakurov[11], A.S. Prokhorov[1,2], V.S. Gorelik[12], L.S. Kadyrov[1], V.V. Uskov[1], R.K. Kremer[13] & M. Dressel[3]

Water is characterized by large molecular electric dipole moments and strong interactions between molecules; however, hydrogen bonds screen the dipole–dipole coupling and suppress the ferroelectric order. The situation changes drastically when water is confined: in this case ordering of the molecular dipoles has been predicted, but never unambiguously detected experimentally. In the present study we place separate $H_2O$ molecules in the structural channels of a beryl single crystal so that they are located far enough to prevent hydrogen bonding, but close enough to keep the dipole–dipole interaction, resulting in incipient ferroelectricity in the water molecular subsystem. We observe a ferroelectric soft mode that causes Curie–Weiss behaviour of the static permittivity, which saturates below 10 K due to quantum fluctuations. The ferroelectricity of water molecules may play a key role in the functioning of biological systems and find applications in fuel and memory cells, light emitters and other nanoscale electronic devices.

[1] Moscow Institute of Physics and Technology, 141701 Dolgoprudny, Moscow Region, Russia. [2] A.M. Prokhorov General Physics Institute, Russian Academy of Sciences, 119991 Moscow, Russia. [3] 1. Physikalisches Institut, Universität Stuttgart, 70569 Stuttgart, Germany. [4] Faculty of Physics, Southern Federal University, 344090 Rostov-on-Don, Russia. [5] Institute of Geology and Mineralogy, Russian Academy of Sciences, 630090 Novosibirsk, Russia. [6] Institute of Physics AS CR, Na Slovance 2, 18221 Praha 8, Czech Republic. [7] Department of Condensed Matter Physics, Faculty of Mathematics and Physics, Charles University, 121 16 Prague 2, Czech Republic. [8] Mathematical Department of The National Research University Higher School of Economics, 101000 Moscow, Russia. [9] Independent University of Moscow, 119002 Moscow, Russia. [10] Institute of Laser Physics, Russian Academy of Sciences, 630090 Novosibirsk, Russia. [11] Kazan Physical-Technical Institute, Russian Academy of Sciences, 420029 Kazan, Russia. [12] P.N. Lebedev Physical Institute, Russian Academy of Sciences, 119991 Moscow, Russia. [13] Max-Planck-Institut für Festkörperforschung, 70569 Stuttgart, Germany. Correspondence and requests for materials should be addressed to M.D. (email: dressel@pi1.physik.uni-stuttgart.de).

Ferroelectric materials exhibit a spontaneous electric polarization, that is, the elementary electric dipoles remain aligned in a certain direction without external electric field; the polarization can be reversed by applying a field. Owing to these unique properties, ferroelectric compounds find a widespread use in microelectronics. The extremely large electric dipole moment makes $H_2O$ molecules the ideal building blocks for ferroelectrics because they enable long-range electrical dipole–dipole interactions. However, there is no firm experimental evidence so far for dipole–dipole ordering within the subsystem of water molecules. The existence of an ordered state of water molecules has been the subject of debate for decades[1–3]. In liquid water, ordering does not occur since the shorter-range H-bonds overwhelm the coupling of the dipoles and prevent the alignment of dipole moments. Even in the solid state, where the oxygen atoms are arranged into an ordered crystal ice lattice, the protons remain disordered within experimentally accessible timescales[1,4], thus leading to zero net macroscopic electrical polarization. Proton ordering in cubic and hexagonal ice XI could lower the ground state energy[5,6], but such ordering was never reliably realized in the laboratory or convincingly observed in outer space[2,7–10].

More than a dozen phases of solid water are known in addition to the conventional hexagonal crystalline ice; and there are even more exotic forms of water, such as biological water, interfacial water, surface water or confined water, that often exhibit structural, dynamical or thermodynamical behaviours not commonly observed in the bulk state[4,11–13]. For example, the dimensionality of water can be decreased by confining the molecules to nanosized voids (pores and channels) or bringing them into contact with extended interfaces. A local disruption of the hydrogen-bond network and their rearrangement into new spatial configurations can, in particular, strengthen the effect of intermolecular dipole–dipole interactions and create conditions favourable for aligning the water dipoles. This type of confined-water ferroelectricity is believed to play a significant role in various phenomena and areas of natural sciences (for example, geology, mineralogy, meteorology, soil chemistry, biology, pharmaceutics, food industry and materials science), including the living organisms (water in cells and membrane channels, and proteins hydration shells[14–17]) and even in the universe (formation of planets or of prebiotic compounds[7]).

Various possibilities for the emergence of confined-water ferroelectricity have been analysed with the help of theoretical studies and computer simulations. Water molecules are predicted to undergo numerous ferroelectric orderings, either when forming nanometre-thick ice layers on two-dimensional (2D) substrates or when confined in nanoscale spaces, such as carbon nanotubes (see, for example, refs 18,19). In these cases the suppression of dipolar interactions by the hydrogen bonds is predicted to be weakened by their reorientation to some external surface, closed or open. As far as the experimental realization of the dipolar ordering is concerned, the situation is not so unambiguous and clear. On the one hand, there seem to be firm indications towards ordered (ferroelectric or antiferroelectric) arrangements of the water molecules within the one-dimensional (1D) channels of carbon nanotubes or molecular organic structures[20,21], or on 2D surfaces[9,22,23]. On the other hand, either the fraction of the polarized dipoles is very low, of the order of 1% or even smaller[22], limited to only few surface layers, or the reliability of the obtained results is put under discussion[8,19].

One could imagine another kind of system that would favour ferroelectricity; one where the intermolecular hydrogen couplings are not reoriented by surface effects but are strongly weakened or completely absent. This can be realized by arranging separate water molecules into a certain matrix in such a way that the spacing $r$ between the dipoles is sufficiently large to prevent short-range hydrogen bonds (interaction length 1–3 Å) but short enough to maintain long-range electric dipole–dipole interaction (interaction length 10–100 Å); such arrangement would favour the alignment of the dipole moments, that is, (anti)ferroelectricity.

In this Communication we present our study of the dielectric response of $H_2O$ molecules embedded into the matrix of the beryl crystal lattice with an intermolecular distance of 5–10 Å. We discover clear signatures of incipient ferroelectricity within the ensemble: a ferroelectric soft mode develops at terahertz frequencies resulting in the Curie–Weiss dependence of the dielectric permittivity that saturates at the lowest temperatures due to quantum effects.

## Results

**Material.** The beryl crystals belong to the gemstone family with the chemical formula $Be_3Al_2Si_6O_{18}$. These are naturally or artificially grown hexagonal crystals (space group $P6/mcc$) that appear in different colour variants depending on the doping by impurity ions. Six-membered rings of $SiO_4$ tetrahedra stacked along the hexagonal $c$ axis form relatively large open channels, whose cross-section is modulated by bottlenecks of $\sim 2.8$ Å separating cages of 5.1 Å in diameter[24] (Fig. 1a,b). Crystals grown in an aqueous environment contain water trapped in the framework of the crystal lattice in such a way that single $H_2O$ molecules reside within the cages; the molecules are trapped in the cages during the growth process, that is, when the cages are formed, and may diffuse to neighbouring vacant sites[25]. The $H_2O$ molecules in the beryl occur in two distinct orientations: either the two protons are aligned parallel to the $c$ axis and the electric dipole points perpendicular to the $c$-direction (type I), or the dipole moment points parallel to the $c$ axis (type II)[26,27]. These two orientations of the water molecules are unambiguously verified in our crystals by observing the $H_2O$ intramolecular vibrational modes $v_1$, $v_2$ and $v_3$ that couple to the polarized radiation strictly differently for the two orientations of the molecule, see Fig. 3 in ref. 28. The type II water molecules are relatively free to rotate around the $c$ axis within the cavities[25,29,30]. In this simple case, only weak van der Waals interaction between water molecules and crystal framework remains[26,31]; the hexagonal symmetry imposes a six-well potential to the type I $H_2O$ molecules[32] as schematically shown in Fig. 1c by the dark green curve.

Let us estimate whether the water molecules, as sketched in Fig. 1c, can actually develop a long-range dipolar order. The dipole–dipole interaction energy of two molecules can be expressed as $U_{d-d} = p^2 r^{-3}$, where $p = 1.85$ D are the electric dipole moments[33] separated by the distance $r$. In the case they are located in neighbouring cages within the same channel at the distance $r = 4.6$ Å, one obtains $U_{d-d} \approx 22$ meV, corresponding to a temperature $U_{d-d}/k_B \approx 260$ K ($k_B$ denotes the Boltzmann constant). For nearest molecules in adjacent channels, $r = 9.2$ Å and $U_{d-d} \approx 3$ meV (corresponding to 30 K). Here the screening of the dipolar interaction by the crystal framework is neglected. The concentration of water in beryl crystals can be determined by chemical analyses. The specimens used in the present study have a concentration of 0.3 $H_2O$ per formula unit, mostly of type I with the concentration of type II molecules not exceeding a few percent (Methods). The molecules are statistically distributed over the cages with more than half of them in strands of two, three and even four molecules. On the basis of these simple estimates and

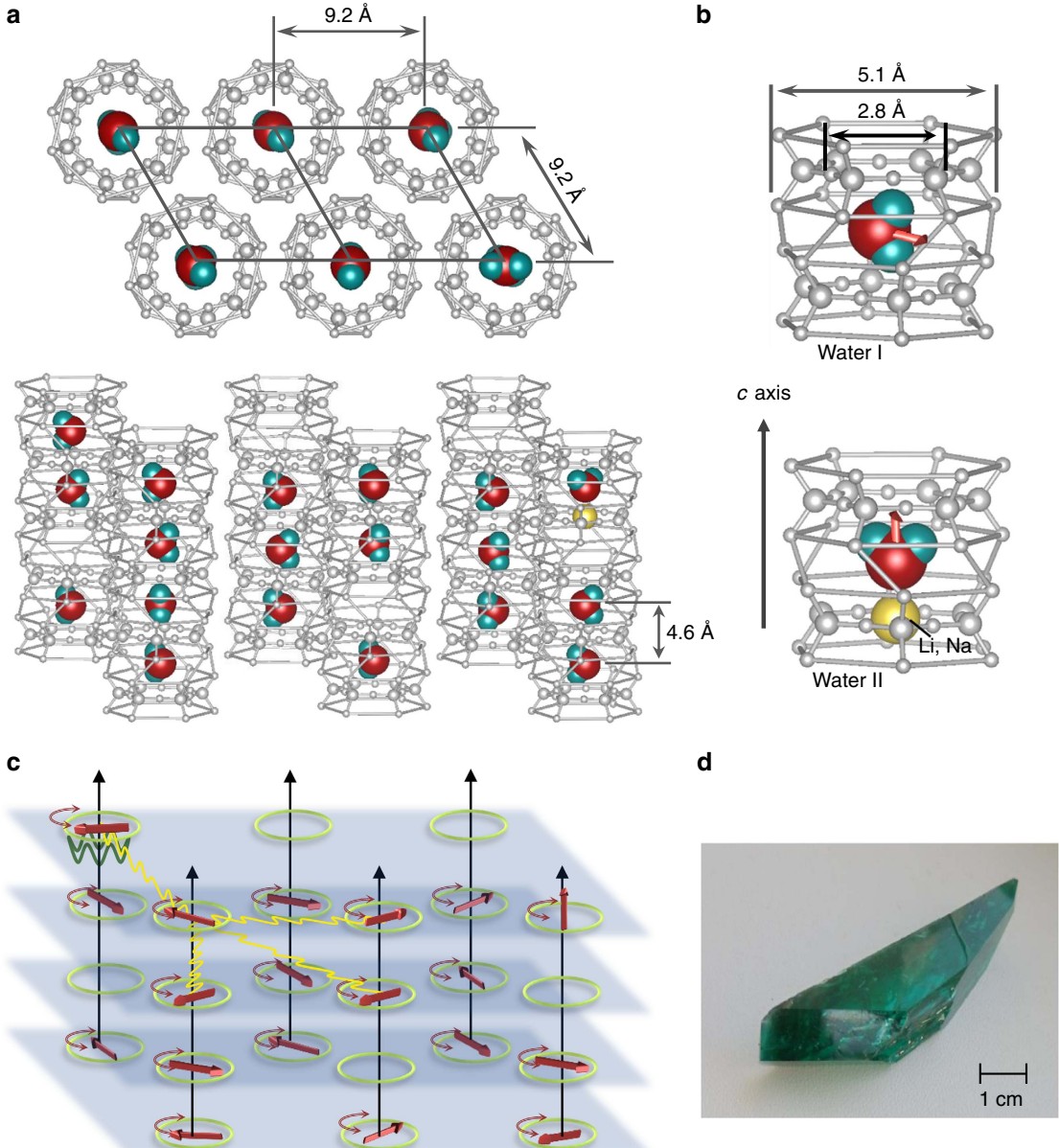

**Figure 1 | Schematic view of water molecules in nano-sized cages of the beryl crystal lattice.** (**a**) Water molecules confined in the channels within the beryl crystal lattice. Three-dimensional and top views with the crystal plotted greyish and the water molecules coloured (oxygen red and hydrogen cyan). The 1D channels are arranged in a hexagonal fashion with 9.2 Å distance and contain cages in a distance of 4.6 Å. (**b**) Water molecules located within structural voids formed by lattice ions. The cages (diameter 5.1 Å) are separated by narrower bottlenecks (2.8 Å). Molecules of type I have their dipole moments (red arrows) perpendicular to the crystallographic *c* axis with the plane of $H_2O$ molecules parallel to *c*; they can perform hindered rotations around the *c* axis experiencing a six-well potential (depth A) due to the hexagonal crystal symmetry. Type II water molecules are turned by 90° relative to those of type I due to Coulomb interactions with alkali ions (Li and Na, shown in yellow) blocking the bottleneck; their dipole moments are directed along the *c* axis. (**c**) Dipole moments of type I molecules. The moments can rotate within the planes perpendicular to the *c* axis. The dipole–dipole interactions (yellow wavy lines) act between the molecular dipoles within the channels where molecular doublets, triplets and so on are formed; the interactions between dipoles in adjacent channels are much weaker owing to their greater mutual distances. (**d**) Photograph of the typical studied beryl crystal.

considerations, ferroelectricity due to alignment of type I water molecules can be expected at temperatures comparable to room temperature.

To search for signs of ferroelectric ordering of water molecules experimentally, we have grown several high-quality beryl single crystals using hydrothermal techniques described in the Methods section. With the help of broad-band dielectric spectroscopy, we looked for characteristic fingerprints of the ferroelectric phase due to ordering of rotatable type I molecules; in particular, we compared the dielectric response corresponding to the electric-field vector **E** of the probing radiation within the plane of molecular rotation, that is, $\mathbf{E} \perp c$, with that for the $\mathbf{E} \| c$ polarization. Several spectrometers enabled us to record the complex dielectric permittivity $\varepsilon = \varepsilon' + i\varepsilon''$ in the frequency range from $v = 1$ Hz up to 2.5 THz at temperatures between 0.3 and 800 K. More details are given in the Methods section below.

**Dielectric and optical properties**. Figure 2 displays the temperature dependence of the dielectric permittivity measured in

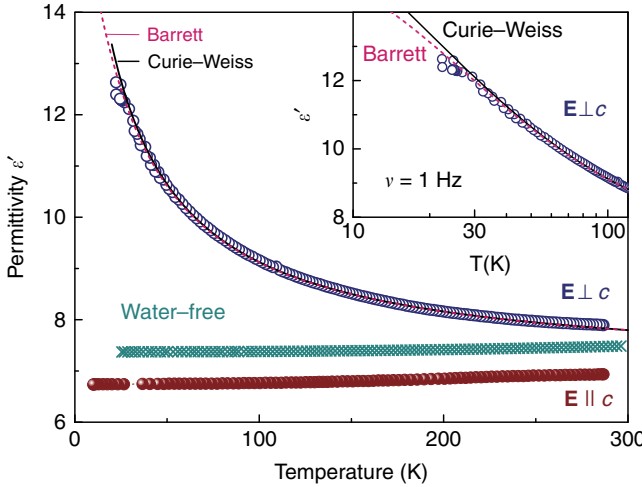

**Figure 2 | Dielectric permittivity of a beryl crystal.** Temperature dependence of the dielectric permittivity of a hydrated beryl crystal at low frequencies ($v = 1\,Hz$) for $\mathbf{E}\perp c$ (blue symbols) and $\mathbf{E}\|c$ (brown symbols). For comparison, the data for a dehydrated crystal in the polarization $\mathbf{E}\perp c$ (green symbols) are also plotted. The inset enlarges the low-temperature range in a logarithmic temperature scale. The solid and dashed lines correspond to fits by Curie–Weiss (equation (1)) and Barrett equations (equation (11)), respectively. The Curie–Weiss parameters are $T_C = -20\,K$, $C = 255\,K$, and the Barrett fit parameters are $T_C = -20\,K$, $T_1 = 20\,K$, $C = 255\,K$; the temperature-independent contribution to the permittivity from higher-frequency excitations is $\varepsilon_\infty = 7$.

the quasi-static limit for the polarization $\mathbf{E}\perp c$, compared with analogous data from a water-free sample and with the $\mathbf{E}\|c$ response. In the geometry of electric field within the plane of $H_2O$ molecular rotation, $\mathbf{E}\perp c$, the permittivity can be fitted well (solid lines) by the Curie–Weiss law[34]

$$\varepsilon'(T) = \varepsilon_\infty + C(T - T_C)^{-1} \qquad (1)$$

where $C$ is the Curie constant, $T_C$ is the Curie–Weiss temperature and $\varepsilon_\infty$ is the temperature-independent contribution to the permittivity from higher-frequency excitations. For $\mathbf{E}\|c$ and for the water-free sample, $\varepsilon'(T)$ is almost temperature independent.

The measurements of the terahertz (THz) electrodynamic response presented in Fig. 3 reveal a broad excitation whose peak frequency $v_0$ gradually decreases as the temperature is reduced; it is this softening of the excitation and the corresponding increase in its dielectric contribution $\Delta\varepsilon = f v_0^{-2}$ that cause the Curie–Weiss increase of $\varepsilon'(T)$ seen in Fig. 2 ($f$ is the oscillator strength of the excitation). In addition to this soft excitation, below about 70 K sharper resonances start to develop in a narrow frequency range 1.2–1.5 THz. These resonances are connected with the wagging-like modes of the $H_2O$ molecules that involve librations of the oxygen ions. No traces of the THz soft mode and of its narrower satellites are present for the polarization direction $\mathbf{E}\|c$, as demonstrated in Fig. 4c; below $50\,cm^{-1}$ only weakly temperature-dependent tails of higher-frequency resonances appear. The comparison of the low-frequency and THz dielectric response of water-free beryl with those of crystals containing $H_2O$ molecules (Fig. 4a,b) makes it clear that it is not the gemstone crystal lattice but the water confined in the channels, which is responsible for the observed behaviour. Since the soft THz mode is only present in the $\mathbf{E}\perp c$ polarization, our experiments undoubtedly probe the dynamics of interacting type I water dipoles as sketched in Fig. 1c. The Curie–Weiss dependence of the dielectric permittivity $\varepsilon'(T)$ and the soft mode in the dielectric spectra are unambiguous signs of a paraelectric behaviour

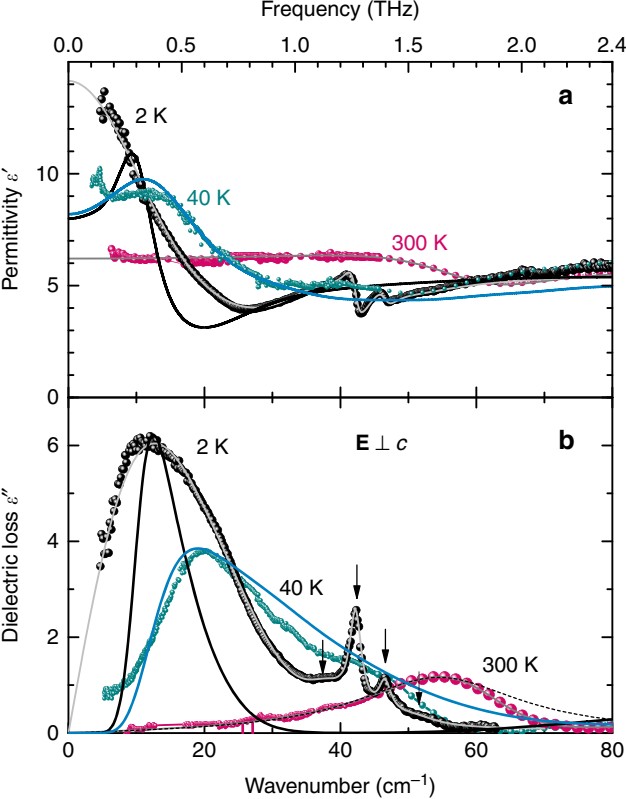

**Figure 3 | Temperature evolution of the dielectric constant and losses.** (**a,b**) Terahertz spectra of the real and imaginary parts of the dielectric permittivity (dots) of a hydrated beryl crystal measured for $\mathbf{E}\perp c$ at different temperatures as indicated. The grey solid lines are fits to the data by a sum of equation (3) (coupled oscillators) for the broad soft mode and equation (2) (damped Lorentzians) for the narrow resonances above $40\,cm^{-1}$, shown by arrows in **b**. The dashed line illustrates a fit with a single Lorentzian term of the spectra at $T = 300\,K$. Black and blue solid lines show the fits according to the six-well librator-rotator model, as described in the text.

expressing the ability of elementary dipole moments to become polarized by the external electric field; this is a typical precursor of a possible phase transition into a state where the dipole moments are macroscopically aligned, leading to ferro- or antiferroelectricity[34]. We note that the centrosymmetric crystal structure of beryl does not exclude the emergence of incipient ferroelectricity of the water subsystem. Even in crystalline ferroelectrics, the incipient ferroelectricity is not connected with any symmetry change since it is only a tendency to a ferroelectric phase transition. In the formed ferroelectric phases there is a spontaneous polarization $P_s$, which breaks the inversion symmetry; in incipient ferroelectrics there is no $P_s$. The best examples of this behaviour are the classical incipient ferroelectrics, strontium titanate $SrTiO_3$ and potassium tantalate $KTaO_3$, which are simple cubic perovskites[34].

**Modelling**. To obtain information on the collective dynamics of the confined water dipoles, a dispersion analysis of the measured spectra was performed; the fits to spectra at different temperatures are demonstrated in Fig. 3. The narrow resonances that develop at low temperatures can be described by a sum of Lorentz oscillators:

$$\varepsilon(v) = \sum_j \frac{f_j}{v_j^2 - v^2 + iv\gamma_j} \qquad (2)$$

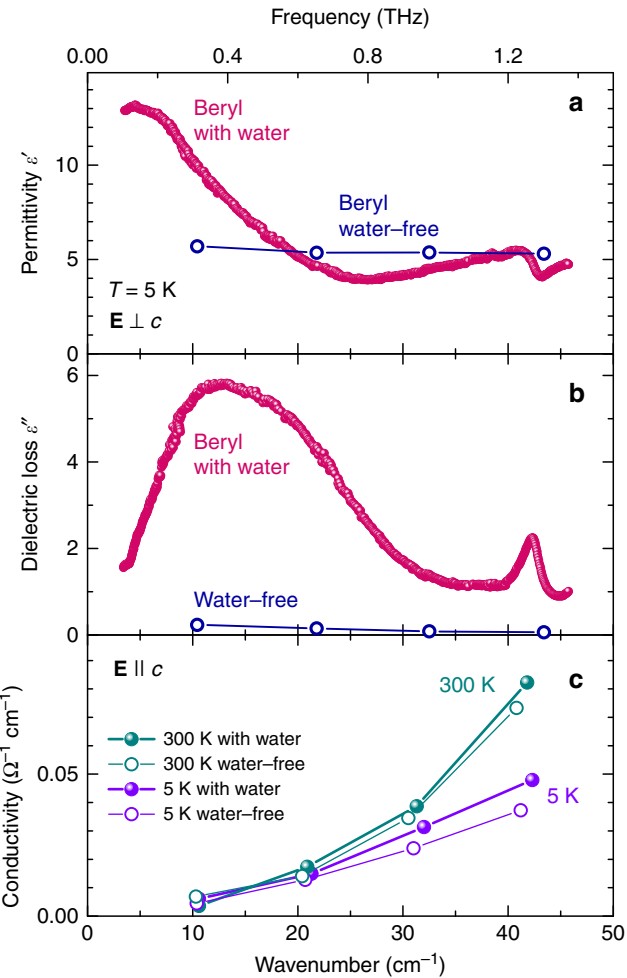

**Figure 4 | Terahertz spectra of hydrated and water-free beryl single crystals.** (**a**,**b**) Spectra of the real and imaginary parts of the dielectric permittivity measured in the polarization $\mathbf{E} \perp c$ at low temperatures, $T = 5$ K. (**c**) Spectra of the optical conductivity of hydrated (cyan) and water-free (violet) crystals measured at $T = 300$ and $5$ K for the polarization $\mathbf{E} \| c$.

where $f_j = \Delta\varepsilon_j v_j^2$ is the oscillator strength of the $j$-th mode, $v_j$ is its resonance frequency, $\Delta\varepsilon_j$ is its dielectric contribution and $\gamma_j$ is the damping constant. The soft excitation, however, has a pronounced asymmetric line shape that cannot be satisfactorily described neither by Lorentzian (dashed line in Fig. 3b) nor by Gaussian profiles. Nevertheless, at all temperatures its spectral shape can be reproduced by the model of two coupled damped harmonic oscillators[35], as demonstrated by the grey solid lines in Fig. 3, where the bilinear coupling is active between modes of the same symmetry. The corresponding complex dielectric permittivity can be written as

$$\varepsilon(v) = \frac{f_1(v_2^2 - v^2 + iv\gamma_2) + f_2(v_1^2 - v^2 + iv\gamma_1) - 2\sqrt{f_1 f_2}(\alpha + iv\delta)}{(v_1^2 - v^2 + iv\gamma_1)(v_2^2 - v^2 + iv\gamma_2) - (\alpha + iv\delta)^2} \quad (3)$$

where $\alpha$ and $\delta$ are the real and imaginary parts of the coupling constant, respectively. The temperature dependences of the fitting parameters of the two components b1 and b2 of the soft excitation are summarized in Fig. 5; here we simply assume a real, temperature-independent coupling constant, $\alpha \approx -2,100$ ($\delta = 0$). Then, only the lower-frequency component b1 exhibits the behaviour typical of a ferroelectric soft mode[34,36]: its

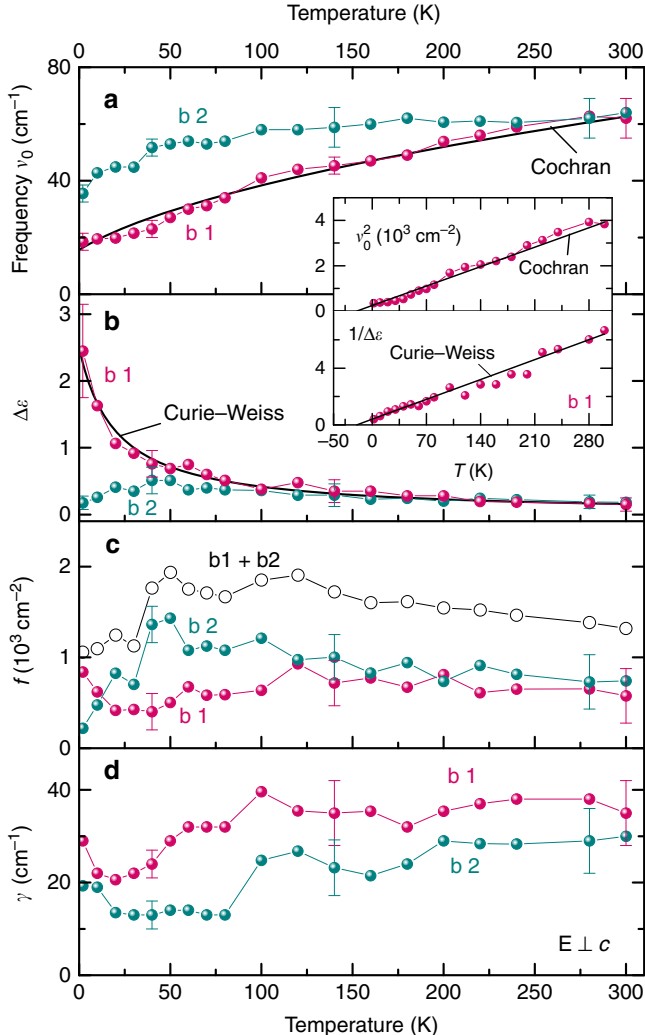

**Figure 5 | Parameters used to fit the soft mode by Curie–Weiss and Cochran models.** (**a**–**d**) Temperature dependences of the parameters of the two components of the soft mode observed in the THz spectra of hydrated beryl crystals for $\mathbf{E} \perp c$: frequencies $v_i$; dielectric contributions $\Delta\varepsilon_i$; oscillator strengths $f_i = \Delta\varepsilon_i v_i^2$ of both components and of their sum (open symbols); and damping constants $\gamma_i$. The two components of the soft mode are labelled by b1 (magenta symbols) and b2 (cyan symbols). Inset: temperature dependences of the squared frequency and inverse permittivity of the component b1 demonstrating the Curie–Weiss, equation (1), and the Cochran, equation (2), behaviours (straight lines). The error bars correspond to the ranges of the data that provide satisfactory description of the original experimental material; that is, fitting the spectra of the complex permittivity and optical conductivity simultaneously.

dielectric contribution follows the Curie–Weiss behaviour $\Delta\varepsilon(T) = C(T - T_C)^{-1}$ and its frequency fulfils the Cochran law[36] characteristic of displacive ferroelectrics

$$v_0 \propto (T - T_0)^{1/2} \quad (4)$$

as shown by the thick lines in Fig. 5. Consequently, we can call this component, b1, the bare ferroelectric soft mode.

**Discussion.** The observed response of interacting type I water molecules is in qualitative agreement with the predictions made for traditional ferroelectric molecular crystals[37–41]. Nakajima and Naya[37] considered interacting dipoles experiencing a double-well

cosine-squared potential of depth $A$ while rotating by 360°. When the thermal energy $k_BT$ significantly exceeds $A$, the collective dynamics of the rotating dipoles (hindered rotations) result in a soft mode in the dielectric spectra whose frequency decreases on cooling. In the limit $k_BT \ll A$, the dielectric response is dominated by restricted rotations (librations) of the dipoles confined in the individual potential wells. At intermediate temperatures both features coexist; as the temperature changes their intensities (spectral weights) vary proportionally to the Boltzmann factor $\exp\{-A/(k_BT)\}$. To describe our results, we have extended the model to the case of six potential wells with the Hamiltonian of the dipoles' movements given by

$$H = H_0 + H_1 \qquad (5)$$

The unperturbed part that includes the crystal field interaction can be written as

$$H_0 = \frac{1}{2I}L^2 - \frac{A}{2}(\cos 6\theta + 1) \qquad (6)$$

The first term stands for the kinetic energy of a dipole with the moment of inertia $I$ and the angular momentum $L$. The second term represents the six-well potential $V(\theta)$ with the wells having the depth $A$ (Fig. 6). The perturbed part $H_1$ in equation (5) describes the dipole–dipole interaction within the mean-field approximation and coincides with the expression (2.4) from ref. 37. On the basis of the fluctuation-dissipation theorem[42] and assuming a linear response, one can connect the microscopic dynamics of the dipoles with their complex polarizability $\phi(\omega) = \phi_1(\omega) + i\phi_2(\omega)$. When applied to our case of hindered rotating dipolar system this yields, for the imaginary part

$$\phi_2(\omega) = \frac{1}{2}\mu^2 \frac{1}{k_BT}\omega \int_{-\infty}^{\infty} dt e^{i\omega t} \langle \cos\theta_0 \cos\theta_t \rangle_0 \qquad (7)$$

Here $\mu$ is the dipole moment, $\theta_t$ is the angular coordinate of the dipole at time $t$, $\theta_0$ is the initial state energy and $\langle \dots \rangle_0$ means the canonical average with respect to the Hamiltonian $H_0$. The real part $\phi_1(\omega)$ can be derived using the Kramers–Kronig relations. The exact solution for the equation of motion $\theta_t$ with the

Hamiltonian $H_0$ in the form of Jacobi functions can be obtained from the elliptic integral

$$t = \sqrt{\frac{I}{2}} \int_{\theta_0}^{\theta_t} d\theta (E + A\cos^2 3\theta)^{-1/2} \qquad (8)$$

here $E$ is the total energy of the dipole. The molecular field approximation allows us to connect the dielectric permittivity of interacting dipoles with the polarizability of a separate dipole; then, regarding $H_1$ as an external perturbation and applying linear response theory[42], we obtain

$$\varepsilon = 1 + 4\pi N \frac{\phi(\omega)}{1 - \gamma\phi(\omega)} \qquad (9)$$

where $N$ is the density of dipoles and $\gamma$ is the molecular field constant. The details of solving the equation of motion in terms of Jacobi functions and of numerical evaluation of the integral (7) are given in ref. 39. Following the described technique, we evaluate the polarizability $\phi(\omega)$ at various temperatures, and using equation (9) we obtain the spectra of the complex dielectric permittivity. For the moment of inertia of the water molecule we use the value $I = 1.022 \cdot 10^{-40}\,\mathrm{g\,cm^2}$ (ref. 4) corrected to the case of rotation axis going through the oxygen atom. The parameters $A$, $N$ and $\gamma$ are varied to achieve the best description of the experimental spectra—the intensity, the position and the half-width of the THz soft mode. The value of the product $\mu^2\gamma$ that determines the coupling of a dipole to the mean field created by all other dipoles is found to be $(1.33 \pm 0.11)$ meV. For the depth of the potential well, the value of $A = (1.41 \pm 0.05)$ meV is obtained, in very good agreement with that (0.83 meV) provided by our density functional theory analysis. The relative fraction of the cages occupied by $H_2O$ molecules, $0.45 \pm 0.13$, (as obtained from $N$) is quite close to the experimentally determined one (about 0.3). With these parameter values, our model provides a reasonable agreement with the experimental spectra (solid lines in Fig. 3). The only discrepancy concerns the soft mode damping (line-width). Whereas the model predicts an increase in the damping with heating, the experimental damping constant is almost temperature independent (Fig. 5d). This implies that actually the width of the soft mode is mainly determined by factors not considered in the model. These can be the following: (i) the presence of impurities or defects, including the alkali ions; (ii) the presence of not only isolated dipoles (lone water molecules) but of doublets, triplets and so on that are randomly spread over the crystal; (iii) the quasi-1D character of the dipole arrangement and, correspondingly, certain distribution in the inter-dipole interaction strength within and across the channels, see Fig. 1c; (iv) possible tunnelling between the potential minima that can act as a source of disorder and thus broaden the line[43,44]; and (v) the temperature dependence of the shape and of the depth of the potential wells and/or of the mean-field coupling constant $\gamma$.

It is instructive to compare the observed dielectric behaviour of the confined water molecules in beryl with the phenomena known from traditional ionic ferroelectrics[34]. In our case the paraelectric behaviour we observe in hydrated beryl is not tied to the lattice but stems from the electric dipoles of labile water molecules enclosed in nanocavities. This particular class of ferroelectricity was predicted for endohedral fullerites[45], where the ordered phase was modelled by a set of polar molecules residing inside stiff $C_{60}$ cages. The present observations provide the first clear experimental evidence of this type of collective state.

Albeit the ordering mechanism is rather different from conventional ferroelectrics, we do observe the typical features indicating paraelectric correlations between water dipoles: (i) the soft mode in the dielectric spectra, (ii) the Curie–Weiss

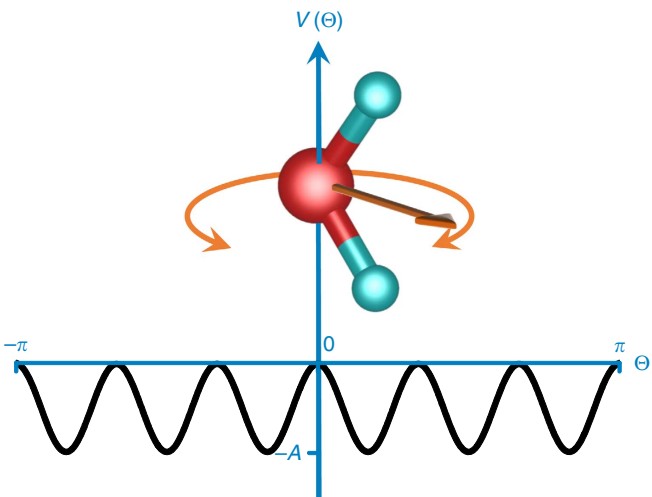

**Figure 6 | Rotation of the water molecule.** Squared cosine potential $V(\theta) = -A \cdot \cos^2 3\theta$ $(A > 0)$ used to model the dynamics of interacting type I water molecules located within the cages of hexagonal beryl crystal lattice. The molecules can rotate around the crystallographic c axis. They rotate freely at temperatures well exceeding the potential energy barriers $(T \gg A/k_B)$, but can only librate within one minima or tunnel between the minima at low temperatures $(T \ll A/k_B)$.

temperature dependence of the dielectric contribution of its bare mode component b1; and (iii) the Cochran temperature dependence of its frequency. Nevertheless, there are distinct differences, such as the two-component character of the soft mode in the whole temperature interval studied. The second component b2 also becomes soft on cooling; however, its dielectric contribution stays temperature independent (Fig. 5b). Consequently, this mode cannot drive a ferroelectric phase transition. Instead, it can be connected to translational vibrations of the $H_2O$ molecule inside the cage. The coupling of the ferroelectric soft mode to other modes is also found in some classical ferroelectrics (see discussions and references in ref. 46) where the bare ferroelectric mode softens and crosses another mode of the same symmetry, resulting in rich and complex dynamics of the coupled vibrations at the so-called anti-crossing temperature[46]. In our case, the anti-crossing point is close to room temperature, as seen from Fig. 5a. Our high-temperature far-infrared transmission data (not shown here) reveal that the frequency of the b2 component does not change significantly up to 1,000 K. The frequency of the bare soft mode b1 increases slightly with temperature; above the maximum occurring around 800–900 K it decreases strongly and is lower by a factor of 2 at $T = 1,000$ K. We associate such a non-monotonic behaviour with structural changes known to occur in the beryl crystal lattice at these temperatures[47,48].

An important question remains: do the observed paraelectric correlations of the water molecules subset result in a real phase transition with macroscopically ordered dipoles? The absence of a divergence in $\varepsilon'(T)$ and a finite $v_0$, observed at temperatures as low as 30 mK, indicate that a real phase transition is not fully reached. This behaviour is known as quantum paraelectricity or incipient ferroelectricity[34,49], most prominently known from $SrTiO_3$ (ref. 50) and $KTaO_3$ (ref. 51). In these crystals, the phase transition is suppressed, mostly, by quantum effects. In the paraelectric phases, on cooling, the temperature dependence of the dielectric permittivity of these compounds first follows the Curie–Weiss behaviour caused by softening of the polar soft modes. At sufficiently low temperatures quantum fluctuations start to play an appreciable role and stabilize the frequency of the soft modes $v_0$, which, in turn, leads to a saturation of $\Delta\varepsilon(T)$ and of the low-frequency $\varepsilon'(T)$. Barrett[52] suggested a particular temperature dependence of these parameters

$$v_0^2 = B\left[\frac{T_1}{2}\coth\left(\frac{T_1}{2T}\right) - T_C\right] \quad (10)$$

$$\varepsilon' = \varepsilon_\infty + \frac{C}{\frac{T_1}{2}\coth\frac{T_1}{2T} - T_C} \quad (11)$$

where $B$ is a constant and the temperature $T_1$ sets the energy range of the quantum effects. The solid lines in Fig. 7 demonstrate that the Barrett formulas perfectly fit the beryl data with a characteristic temperature $T_1 = 20$ K. We note that the same set of parameters describes the temperature variation of permittivity (Fig. 2) and the Curie–Weiss and Cochran behaviours of parameters $\Delta\varepsilon$ and $v_0$ of the bare soft mode b1 (Fig. 5a,b, inset). The slight decrease in the permittivity below 2 K (see the inset of Fig. 7a) can be attributed to an onset of short-range spatial (anti-)ferroelectric correlations.

We can conjecture that the mechanism that suppresses the phase transition may be connected to tunnelling of $H_2O$ molecules between the states in the six-well potential[43,44]. The ferroelectric phase transition may also be inhibited because the structural channels of the beryl crystal are not completely filled by water molecules; together with the weaker interaction among the

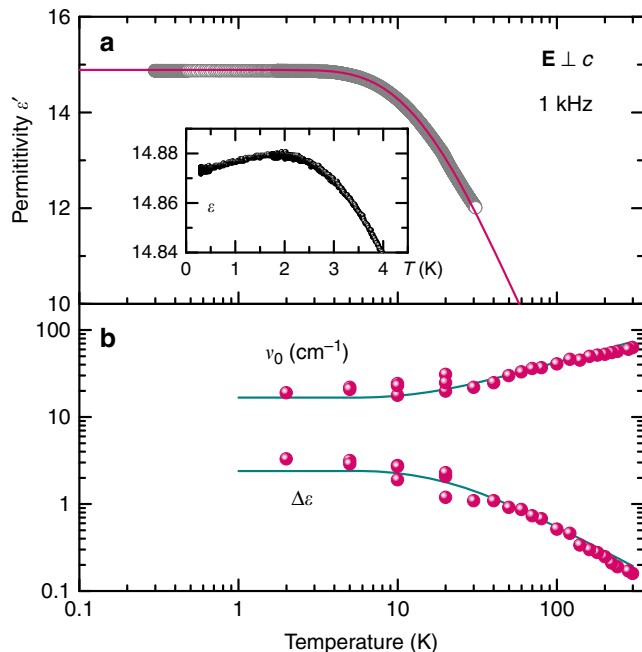

**Figure 7 | Low-temperature behaviour of the soft mode.** (**a**) Temperature-dependent dielectric permittivity of a hydrated beryl crystal measured for $\mathbf{E} \perp c$ at 1 kHz (dots) and its description by the Barrett formula (equation (11)) with the parameters $T_1 = 20$ K, $T_C = -20$ K, $C = 211$ K, $\varepsilon_\infty = 7.9$. Inset: lowest temperature (down to 30 mK) permittivity in an expanded scale. The dependence demonstrates a weak maximum around $T = 2$ K associated with the onset of short-range spatial ferroelectric (antiferroelectric) correlations between the dipole moments. (**b**) Temperature dependences of the dielectric strength $\Delta\varepsilon$ and frequency $v_0$ of the bare soft mode (b1 component of the soft mode). The lines are fits by the Barrett formulas (equations (10) and (11)) with the parameters $T_1 = 20$ K, $T_C = -20$ K, $C = 75$ K, $B = 22$.

$H_2O$ clusters in adjacent channels, this may prevent the formation of a macroscopically ordered state.

Conventional ferroelectrics are distinguished by the possibility of changing the polarization by external parameters, like electric field or pressure[34]. For example, reversing the spontaneous polarization by an external field usually requires field strengths in the order of $10\,\mathrm{kV\,cm^{-1}}$; biaxial pressure induced by the substrate triggers the ferroelectric and antiferrodistortive phase transition in $SrTiO_3$ thin film[53]. In our specimens, an applied field of up to $15\,\mathrm{kV\,cm^{-1}}$ was not sufficient to flip the $H_2O$ dipole moments and to obtain a net macroscopic polarization of the sample. Indeed, a simple estimate of the field that couples together the two water dipoles at a distance $r = 4.6$ Å provides a value $E_{d-d} = pr^{-3} \approx 6,000\,\mathrm{kV\,cm^{-1}}$. This value is typical of crystalline internal fields and exceeds by far the fields needed to shift ions even slightly in regular ferroelectrics.

Very recently Kolesnikov et al. reported inelastic neutron spectroscopy (INS) and density functional theory investigations on the energy states and hydrogen vibrational modes of water molecules confined in natural beryl[54] that directly confirm our earlier suggestion[28,55] about the quantum tunnelling of water molecules within the hexagonal potential. In the spectral range from 0.2 to 20 meV a number of peaks are associated with the transitions between tunnel-split energy levels of the trapped $H_2O$ molecules. With the incident neutrons directed parallel to the $c$ axis, the translational vibrations of water are observed at 11 meV (89 $\mathrm{cm^{-1}}$) previously seen by THz spectroscopy[28,55]; but also several additional tunnelling resonances, not seen by optics, are

revealed due to different selection rules. INS could not detect the narrow resonances at 1.2–1.5 THz (Fig. 3a) as they are due to librations of oxygen ions. The evolution of the INS spectra below 5 meV (Fig. 1a in ref. 54) resembles the soft-mode response discovered in the THz spectra: at $T = 45$ K there is a bump above 2 meV of a width of 2–3 meV that gets narrower and more intense while cooling, similar to what we observe in the THz spectra (Fig. 3b). The energy barrier between the minima of the potential of ~50 meV obtained from INS seems to be rather large.

We conclude that the structural channels of the beryl crystal lattice provide a suitable matrix to achieve an ordered arrangement of individual $H_2O$ molecules. The properties of this unique water state are determined by intermolecular electrical dipole–dipole interactions rather than by the hydrogen bonds as in bulk water or solid ice. We observed clear fingerprints of incipient ferroelectricity of the coupled water molecules, namely the Curie–Weiss temperature dependence of the dielectric constant that transforms into a Barrett-like saturation below 10 K, and the ferroelectric soft mode whose dielectric strength and frequency also level off below 10 K. The structural channels of beryl crystals offer a model system to study, on a molecular level, the properties of quasi-1D chains of confined $H_2O$ molecules coupled via electric dipole interactions. In the future, a deeper insight into the fundamental properties of nano-confined water and, in particular, its ferroelectricity could be obtained by using other crystalline/artificial networks, studying the influence of the cage geometry, their size and topology (1D, 2D and three-dimensional). This could also help to reveal the underlying principles of microscopic water-related properties in various systems, like the transport of charges across biological membranes, role of hydration–water ferroelectricity in functioning of proteins or exotic phenomena in carbon nanotubes. One can also consider analogous manipulations with nanoscopically confined magnetic moments, or molecules possessing both magnetic and electric dipole moments, with a possible nanoscale multiferroicity.

## Methods

**Samples growth and characterization.** High-quality beryl single crystals of centimetre size (Fig. 1d) were grown on a seed according to the hydrothermal growth method[56]. Starting from $SiO_2$, $Al_2O_3$ and $BeO$ oxides the synthesis is performed at the temperature of ~600 °C and under partial water pressure of 1.2–1.5 kbar resulting in a water content of the crystals of about 0.3 per formula unit[57]. To avoid admixture of foreign impurities, the growth process was carried out in hermetically welded gold capsules of ~100 ml volume. The obtained crystals were subjected to a full chemical analysis according to wet chemistry methods. The detection limit was 0.01 wt%, the relative error—0.6%. The results of analysis (Table 1) were recalculated to the crystal chemical formula, in accordance with Bakakin and Belov[58]. The resultant chemical formula then was $(Be_{2.999}Cu_{0.001})_{\Sigma=3}(Al_{1.951}Si_{0.011}Fe_{0.041})_{\Sigma=2.003}(Si_{5.982}Be_{0.018})_{\Sigma=6}$ $O_{18}(Na_{0.009}Li_{0.007})_{\Sigma=0.017}(H_2O)_{\Sigma=0.257}$. The water content in the crystals was considered to be equal to the loss on ignition, as they did not contain any other volatile impurity components or impurities of heavy alkali metals $K^+$, $Rb^+$ and $Cs^+$ that can enter the channels and prevent the removal of water molecules during heating. It is well known (see, for example, refs 26,27,59) that water molecules occupy the channels where they find enough space; implantation of the $H_2O$ molecules in other position(s) would cause noticeable deformation of the crystal lattice for which we have no indication. Besides the performed chemical analysis, the presence of water in our crystals is clearly indicated spectroscopically—by observation of well-known intramolecular modes $v_1$ (stretching mode), $v_2$ (scissor mode) and $v_3$ (asymmetric stretching mode) that are slightly shifted in the crystal relative to the modes positions in free $H_2O$ molecule

$v_1 = 3,656.65$ cm$^{-1}$, $v_2 = 1,594.59$ cm$^{-1}$ and $v_3 = 3,755.79$ cm$^{-1}$; the combined vibration $v_1 + v_2$ is also seen at around 5,300 cm$^{-1}$. Correspondent infrared spectra are not shown in the present paper but typical intramolecular vibration resonances detected in other crystal are presented in our previous publication[28].

Within the crystal, water molecules can form doublets, triplets and so on, disposed in adjacent cages separated by the bottlenecks inside the channels. We can estimate the concentrations of such agglomerations as follows. Let the cluster of $k$ molecules have a length $k$, the total number of water molecules be $M$ and that of cages $N$; $1 \le k \le M$. Combinatorial consideration allows us to derive the following expression for the mathematical expectation $n_k$ of number of clusters that contain $k$ sequential water molecules and that are enclosed at both ends either by lone empty sites or by a sequence thereof

$$n_k = \frac{2C_{N-k-1}^{M-k} + (N-k-1)C_{N-k-2}^{M-k}}{C_N^M}$$
$$= \frac{(N-M+1)(N-M)}{(N-k)} \cdot \frac{(N-k)!\,M!}{(M-k)!\,N!} \quad (12)$$

This precise expression allows simplification to a formula that is more convenient for practical use. Assuming that $M$, $N$, $M-k$ and $N-k$ are large and using the Stirling expression $n! \approx \sqrt{2\pi \cdot n} \cdot (n/e)^n$ we get

$$n_k \approx N\left(1 - \frac{M}{N}\right)^2\left(\frac{M}{N}\right)^2 \quad (13)$$

Let us use the relative fraction $m_k = k \cdot n_k M^{-1}$ of water molecules residing in a clusters of length $k$. With a total concentration of water molecules $C_w = MN^{-1}$ we obtain

$$m_k \approx k \cdot \left(1 - \frac{M}{N}\right)^2 \cdot \left(\frac{M}{N}\right)^{k-1} = k \cdot (1 - C_w)^2 \cdot (C_w)^{k-1} \quad (14)$$

This formula is self-consistent since it fulfils the condition $\sum_{k=1} k \cdot (1 - C_w)^2 (C_w)^{k-1} = 1$. The results of the corresponding calculations are presented in Fig. 8 by solid lines. As an independent check, we have performed numerical computer simulations to find out how $M$ water molecules can be distributed among $N$ empty cavities ($M < N$), at different values of $M$. We assume an array of $N = 10^5$ elements, each of which is assigned a value 0. Using the standard random number generator, an integer number $i \in [1, N]$ was generated and the $i$-th element from those not equal to zero was given a value 1. After each such step the values of $N$ and $M$ deceased by 1. The process was repeated as long as

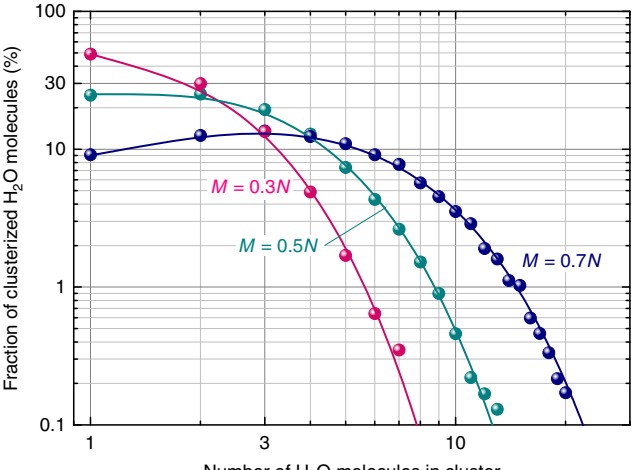

**Figure 8 | Clustering of water molecules.** Fraction of clustered water molecules versus the number of water molecules in a cluster. The calculations are done for different content of water molecules $M$ relative to the number of cages $N$. The red dots and red line ($M = 0.3N$) correspond to the particular crystal studied here. The lines are calculated according to the expression (14), dots correspond to numerical modelling.

| Table 1 | Chemical composition (in wt%) of the investigated beryl crystal. | | | | | | | | | | |
|---|---|---|---|---|---|---|---|---|---|---|---|
| Sample name | SiO$_2$ | Al$_2$O$_3$ | BeO | Fe$_2$O$_3$ | MnO | Cr$_2$O$_3$ | CuO | Li$_2$O | Na$_2$O | LOI* | Total |
| 2934 | 66.12 | 18.27 | 13.86 | 0.60 | 0.00 | 0.00 | 0.01 | 0.02 | 0.05 | 0.85 | 99.78 |
| *LOI, loss on ignition under 1,100 °C during 1.5 h. | | | | | | | | | | | |

the remaining value of $M$ was positive. At the end, the frequencies of occurrence of clusters of varying lengths were determined. The results of such modelling are indicated by dots in Fig. 8. Clearly, both used analyses provide the same results.

**Spectrometers.** For optical measurements a crystal of about a cubic centimetre in size was oriented using X-rays and cut in slices with the crystallographic $c$ axis within or perpendicular to their planes. These geometries allowed measuring the optical response in two principal polarizations with the electrical vector of the probing radiation oriented parallel or perpendicular to the $c$ axis. Dielectric measurements were performed using four spectrometers. At frequencies from about 1 Hz to 1 MHz and temperatures 5 to 300 K the dielectric response was measured in vacuum using a NOVOCONTROL Alpha AN High Performance Frequency Analyzer equipped with a He-flow cryostat JANIS ST-100. The experimental specimens were fabricated as thin plane-parallel polished plates with thicknesses of $\sim 200\,\mu m$. Pt-Au-electrodes (diameter 8 mm) were evaporated using a Bal-Tex SCD 050 sputter coater onto the principal faces of the plates. The contacts for applying the electric field were provided by silver wires fixed to the electrodes by a silver paste. The radiofrequency response at temperatures down to 300 mK was measured at a fixed frequency of 1 kHz using an Andeen-Hagerling 2500A capacitance bridge connected to the cryostat utilizing a single-shot $^3$He insert with embedded coaxial cables. In the terahertz and sub-terahertz ranges the spectra of complex dielectric permittivity and optical conductivity were measured using a spectrometer based on backward-wave oscillators[60] operating at frequencies from 30 GHz to 1.5 THz and in the temperature interval from 2 to 300 K. This is realized by determining the spectra of complex (amplitude and phase) transmission coefficients of plane-parallel samples and subsequent evaluation of optical constants based on the Fresnel equations. In the frequency interval of 0.15–2.5 THz, time-domain THz spectroscopy measurements of complex transmittance were performed using a custom-made spectrometer based on a Ti:sapphire femtosecond laser. The samples were cooled down to liquid helium temperatures in an Optistat (Oxford Instruments) helium-flow optical cryostat with Mylar windows. For infrared measurements, a customized Fourier-transform spectrometer Bruker IFS-113V was used to measure the spectra of reflection and transmission coefficients employing a KONTI bath cryostat (CryoVac). The reflectivity spectra were recorded using samples with a thickness of about 1 mm and for the transmissivity measurements thinner (about $100\,\mu m$) samples were used. To identify water-related features in the measured spectra we performed dielectric and optical experiments on dehydrated samples. The dehydration procedure included keeping the crystals heated up to $1,000\,°C$ in vacuum for 24 h. A comparative analysis of optical spectra of samples with and without water allowed us to unambiguously distinguish water-related absorption resonances from those due to phonons or impurities.

**Data availability.** The data that support the findings of this study are available from the corresponding authors on request.

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

## Acknowledgements

We acknowledge fruitful discussions with V.V. Lebedev, S. Tretyak, A. Zhugayevich, A.A. Pronin, A.V. Pronin, A. Mukhin, G.A. Komandin, D. Efremov, V. Železný, S. Kamba, J. Hlinka and B. Gompf. The work was supported by the Russian Ministry of Education and Science (Program '5top100'), by RFBR project 14-02-00255 and by the Czech Science Foundation (project 14-25639S). Dielectric experiments in the mK range were performed at the Magnetism and Low-Temperature Laboratories (MLTL), supported within the programme of Czech Research Infrastructures (project no. LM2011025). We acknowledge funding of the Deutsche Forschungsgemeinschaft (DFG). We thank G. Untereiner, V. Skoromets and A.E. Sashin for their experimental help and assistance.

## Author contributions

E.S.Z., L.S.K., V.V.U., C.K and F.K. carried out the THz experiments; M.S. and J.P. carried out the radiofrequency experiments; T.O. carried out the infrared experiments; E.S.Z., C.K., F.K. and L.S.K. analysed the data; V.G.T., E.V.P., D.A.F. and G.S.S. prepared the samples; M.A.B., V.I.T., E.S.Z., P.V.T., J.P., A.S.P., V.S.G. and R.K. carried out the theoretical analysis and the numerical simulations; B.P.G. and M.D. conceived and supervised the work; all authors contributed to the manuscript.
