## [Peer Review File · Nature Communications]

Reviewers' comments:

Reviewer #1 (Remarks to the Author):

The authors state that they observed incipient ferroelectricity in beryl due to chains of water dipole-dipole interaction along crystal c-axis. Beryl has a centrosymmetric crystallographic structure, and it is known, that "it is a necessary, but not sufficient, condition for ferroelectricity that the crystal lack a center of symmetry" (see C. Kittel, Introduction to Solid State Physics, John Willey & Sons, Inc., 1963, p. 182). Also, R.M. Levy in his book Principles of Solid State Physics, Academic Press, 1974, p.186, wrote "to be a ferroelectric, the point group must contain a unique nonequivalent direction. ... Fig. 5.23 illustrates these conditions with three point-group diagrams. In (a), the beryl structure, there is a center of symmetry at P so this point group is not piezoelectric or ferroelectric." Therefore, the crystallographic symmetry of beryl is not compatible with the ferroelectric order.

A recent theoretical paper [8] on possibility of ferroelectric ice-XI stated, that "the depolarization field ... dominates the energetics, making the existence of ferroelectric ice unlikely." The authors have estimated that the depolarization field in ice-XI should be $E = 7.24 \times 10^9$ V/m, which is an electric field that cannot be supported by any material: even a small crystal ... will develop enormous depolarization fields." Hence, this quantitative estimate shows that ferroelectric state of water in beryl is practically impossible.

Other comments:

- 1) The orientation of water I in beryl shown in Fig. 1 is incorrect; the water H-H direction should be almost parallel to the crystal c-axis, while it is shown as being in ab-plane.
- 2) Fig. 5 shows that the value of the soft-mode is about 20-60 cm^{-1} or 2.5-7 meV. It is unclear, how can these soft-modes be realized in the potential well which has the depth of $A=1.41$ meV (line 244).
- 3) The caption to the Fig.6 says, "water-I molecules ... can rotate around the axis passing through oxygen ion within the plane perpendicular to the crystallographic c-axis." This is a wrong statement, which disagrees with the known structure of water I molecules in beryl.

I do not think that the manuscript meets the criteria for publication in Physical Review Letters.

Reviewer #2 (Remarks to the Author):

The manuscript examines the ferroelectricity of water molecules confined to nano-channels of beryl through experiments. To avoid formation of hydrogen bonds between water molecules, the authors had experimentally placed water molecules in the channels of a beryl crystal, and the incipient ferroelectricity is observed. The authors attributed the ferroelectricity of confined water molecules to the disruption of hydrogen bond network and the intermolecular dipole-dipole interactions. This work is of great interest and important. I support publication of this work in Nature Communications after the authors address the following comments:

Experimental evidence of placing water molecules in the structural channels of a beryl crystal must be provided in the supporting information.

Discussion on how to put water molecules into the nano-channels of beryl since the bottleneck is very narrow (~ 2.8 Å).

Related experimental study could be cited (e.g. PNAS 108, 3481 (2011)).

The orientation profile of water molecules confined to nano-channels should be provided.

Reviewer #3 (Remarks to the Author):

The authors in this work report incipient ferroelectricity of water molecules confined to nano-channels of beryl crystals, in which the intermolecular hydrogen coupling is strongly weakened or completely absent. Based on the theoretical and experimental investigation on the property of the water, the authors reveal that it is the intermolecular electrical dipole-dipole interactions, rather than by the hydrogen bonds as in bulk water or solid ice, plays a key contribution to the incipient ferroelectricity of the confined water. Hence, the work provides new insight in the ferroelectricity of the water. Owing to the ferroelectricity of confined water playing a significant role in various phenomena and areas of natural sciences, I'd like to recommend publication of the work in nature communications after major revision.

- 1) The authors claim in the work that "there is no firm experimental evidence so far for ordering within the subsystem of water molecules". In fact, there are some examples experimentally demonstrated existence of ferroelectric and antiferroelectric water in confined system. Because both are related to the order of water molecules, thus the claim should be revised and the related references should be cited.
- 2) How about the crystal size? I'd like to suggest the authors provide the picture of the crystal.
- 3) From a statistical point of view, there is no doubt that each cage in the compound contains 0.3 water molecules. The actual situation may be that no water exists in part of the cage, since the water molecules can form doublets, triplets, etc., disposed in adjacent cages separated by the bottlenecks inside the channels as mentioned by the authors. Considered that the diameter of the cage is about 5.1 Å, is it possible to encapsulate one water molecule into the cage. If possible, I'd like to suggest the author to prepare the compound and investigate the property of the confined water.

Reviewers' comments:

Reviewer #1 (Remarks to the Author):

I would like to thank authors for response to my comments, and I accept some of them. Nevertheless, I still have a question regarding the observed soft modes at about 20 and 40 cm^{-1} (2.5-5 meV) at 5 K, which increase their values to $\sim 60 \text{ cm}^{-1}$ (7.5 meV) at 300 K. In addition at low temperatures the authors observed sharp peaks in the range 1.2-1.5 THz (4.8-6 meV) which were explained as twist-like modes of the H₂O molecules. Recently, a paper on inelastic neutron scattering (INS) study of water in beryl has been published (Kolesnikov et al., Phys. Rev. Lett. 116 (2016) 167802), which presented observation of multiple tunneling peaks in the INS spectra in the energy range 2-118 cm^{-1} (0.27-14.7 meV) at low temperatures, which strongly decrease their intensity with temperature increase, but without noticeable change in the peaks positions. Unfortunately, the peaks observed in that paper are not visible in the manuscript under review, and the peaks in the manuscript at 20-40 cm^{-1} are completely missed in the INS spectra. Another related comment is regarding the estimated depth ($A=1.41 \text{ meV}$) for the water molecule rotational potential well. In the above mentioned paper on INS study of water in beryl the observed maximum splitting of the ground state is 14.7 meV, that means that the depth of the potential well should be larger than that value, so the value $A=1.41 \text{ meV}$ in the manuscript is at least one magnitude order smaller, than obtained from INS study. It is well known that the INS spectra are very sensitive to the vibrational modes of hydrogen due to anomalously large neutron scattering cross-section on hydrogen, and INS spectra directly related to the density of vibrational states.

Based on these, it should be clarified or discussed in the manuscript why the observed soft modes (as well as twist-like modes) are completely missed in the INS spectra of water in beryl, and why there is a big discrepancy on the value of depth of the potential well. Could it be that the peaks observed in the manuscript are not related to the vibrational modes of water in beryl? Maybe water influences the beryl cage vibrations, which were observed in the manuscript? Therefore, I need to have answers for these questions, before I can make the recommendation for the publication.

Reviewer #2 (Remarks to the Author):

The revisions are satisfactory.
I'd like to recommend publication in Nature Commun.

Reviewer #3 (Remarks to the Author):

I am satisfied with the revised manuscript and recommend publication of the work.

REVIEWERS' COMMENTS:

Reviewer #1 (Remarks to the Author):

I thank the authors for the answers and the manuscript modification, but I still disagree with some of their statements.

Regarding the observed resonances at 1.2-1.5 THz, which "are connected with the twist-like modes of the H₂O molecules that involve librations of the oxygen ions [32]." Twist-like mode of water molecule is a libration of water molecule around its dipole moment (see e.g. https://en.wikipedia.org/wiki/Molecular_vibration), therefore this mode should be inactive in optical spectroscopy since it does not affect the dipole moment of the molecule. Also, the twist mode does not involve vibration of oxygen at all. These resonances could be, hypothetically, due to a wagging mode (librations of water-I molecule around the axis of channels / c-axis of beryl). In this case the 1.5 THz (6 meV) excitation would require the potential depth (A) to be larger than this value, which disagrees with the estimates made in the manuscript, A=1.41 meV. In addition, eigenvectors of water librational modes are much larger for hydrogen than for oxygen, therefore INS spectra for water librational modes are mostly originated due to neutron scattering on hydrogen, and these modes are the most strong in the water INS spectra.

The above remarks should be responded before I can recommend the manuscript for the publication in Nature Communications.

Replies to remarks of the First Reviewer

Reviewer 1:

The authors state that they observed incipient ferroelectricity in beryl due to chains of water dipole-dipole interaction along crystal c-axis. Beryl has a centrosymmetric crystallographic structure, and it is known, that "it is a necessary, but not sufficient, condition for ferroelectricity that the crystal lack a center of symmetry" (see C. Kittel, Introduction to Solid State Physics, John Wiley & Sons, Inc., 1963, p. 182). Also, R.M. Levy in his book Principles of Solid State Physics, Academic Press, 1974, p.186, wrote "to be a ferroelectric, the point group must contain a unique nonequivalent direction. ... Fig. 5.23 illustrates these conditions with three point-group diagrams. In (a), the beryl structure, there is a center of symmetry at P so this point group is not piezoelectric or ferroelectric." Therefore, the crystallographic symmetry of beryl is not compatible with the ferroelectric order.

Our reply.

It is very important to distinguish between ferroelectricity and incipient ferroelectricity. Of course, beryl crystal is centrosymmetric, so it cannot be ferroelectric; but **incipient** ferroelectricity is fully compatible with centrosymmetry - see the classical incipient ferroelectrics strontium titanate SrTiO_3 [Barker and Tinkham, Physical Review 125, 1527 (1962)] and potassium tantalate KTaO_3 [Samara, J. Phys. Cond. Matt. 15, R367 (2003)], which are simple cubic perovskites. In contrast to a true ferroelectric state where the microscopic polarization is linked to a structural distortion with a translational symmetry, the incipient ferroelectricity in beryl corresponds to transient alignments of the water molecules due to their mutual interactions. Consequently, the earlier symmetry analyses of beryl, performed by X-ray diffraction, inherently neglect the tiny time-dependent structural deformations due to the fast moving water molecules. Note that the average long-range symmetry of the water-containing and water-free beryl is certainly the same within the accuracy of XRD. It cannot differ since the water molecules are dynamically disordered and therefore their effect on the symmetry must be nil in the time average.

We have added to the manuscript (p. 6/7) a few sentences clarifying this issue to readers who are not well familiar with the incipient ferroelectricity.

Reviewer 1:

A recent theoretical paper [8] on possibility of ferroelectric ice-XI stated, that "the depolarization field ... dominates the energetics, making the existence of ferroelectric ice unlikely." The authors have estimated that the depolarization field in ice-XI should be " $E = 7.24 \times 10^9$ V/m, which is an electric field that cannot be supported by any material: even a small crystal ... will develop enormous depolarization fields." Hence, this quantitative estimate shows that ferroelectric state of water in beryl is practically impossible.

Our reply.

The suppression of the ferroelectric order by depolarization fields is crucial for conventional ferroelectric thin films or small particles (submicrometer sizes), because the origin of the depolarization fields is closely connected to the geometry of the object. This field comes from unscreened charges that are generated at the surfaces or interfaces. The effect is nicely treated by P. Parkkinen et al. [8] with regard to water ice XI. The corresponding structure considered in Ref. [8] is essentially two-dimensional and it is shown that the ferroelectric alignment of the water dipoles in the ice slab will be suppressed by a depolarization field amounting to 7.24×10^9 V/cm, as correctly stated by the Reviewer. However, this large field is generated by a potential drop of 4.64 V across a very thin (6.54 Angstroms) slab of ice XI. Obviously, the geometrical arrangement of partly disordered (single, dimers, trimers, etc.) water molecules in the channels

of beryl has nothing in common with the two-dimensional water molecular arrangement in ice XI and, correspondingly, the above estimate of the depolarization field cannot be applied to the case of water-containing beryl.

Furthermore, as we write two paragraphs above, we are considering incipient ferroelectricity, whereas Ref. [8] denies the existence of true ferroelectricity with a quasi-static polarization. In contrast, in the case of incipient ferroelectric dynamics, the dipolar field oscillates with the soft mode (proton hopping) frequency (a few THz) and it cannot be fully compensated by the beryl lattice, whose inertia is much higher. The dipolar field might be partly compensated along the channels by some defect charges, if they were available, but the dynamics of the defects are much slower, so this cannot completely suppress the transient alignments of the water molecules. It could somewhat stiffen the measured soft mode frequency and decrease the extrapolated negative Curie temperature. This was not considered in our model calculations (it can hardly be included), but it can help to understand why the soft-mode energy is somewhat higher than the hopping barrier in Fig. 6 in our calculations, which answers partly also the 2nd additional question of the Reviewer, see below.

Reviewer 1:

Other comments:

1) The orientation of water I in beryl shown in Fig. 1 is incorrect; the water H-H direction should be almost parallel to the crystal c-axis, while it is shown as being in ab-plane.

Our reply.

The Reviewer is right, this was an error in preparation of the figure. We have corrected the drawing in the revised version. We thank the Reviewer for pointing it out.

2) Fig. 5 shows that the value of the soft-mode is about 20-60 cm⁻¹ or 2.5-7 meV. It is unclear, how can these soft-modes be realized in the potential well which has the depth of A=1.41 meV (line 244).

Our reply.

We are dealing with the system of *coupled* dipoles *rotating above* the 6-well potential of depth A, and this potential makes some kind of friction for the dipolar rotation. Clearly, the frequency (energy) of the resultant collective soft mode should depend both on the wells' depth *and* on the coupling strength. It is not straightforward to quantitatively compare the energy value of the soft mode with the value of the wells depth and of the coupling energy. The task was analyzed by Nakajima and Naya [38] for the case of coupled dipoles rotating above the two-well potential. However, their analysis does not allow for a simple quantitative comparison of the above values. We have extended the theory of Nakajima and Naya to the case of six-well potential relief, and after going through all steps of the analytical and numerical analyses, we obtained the value of A=1.41 meV that could best describe the *experimental* spectra. In addition this value is independently confirmed by our *ab initio* DFT analysis of the system "water molecule in beryl" where we have obtained A=0.83 meV, that is quite close to 1.41 meV taking into account the known approximations of the Nakajima/Naya theory when extended to six minima and applied to water in beryl. For these reasons, we are convinced that this objection of the Reviewer does not reflect any incoherence in our paper.

3) The caption to the Fig.6 says, "water-I molecules ... can rotate around the axis passing through oxygen ion within the plane perpendicular to the crystallographic c-axis." This is a wrong statement, which disagrees with the known structure of water I molecules in beryl.

Our reply.

We thank the Reviewer for pointing out the poor wording, that can lead to confusion and misunderstanding. We have changed the figure caption, which now reads in total:

“Fig.6. Squared cosine potential $V(\theta) = -A\cos^2 3\theta (A>0)$ used to model the dynamics of interacting water-I molecules that are located within the cage of hexagonal beryl crystal lattice and can rotate around the crystallographic c -axis. The molecule rotates freely at temperatures well exceeding the potential energy barriers ($T \gg A/k_B T$), but can only librate within one minimum or tunnel between the minima at low temperatures ($T \ll A/k_B T$).”

Replies to remarks of the Second Reviewer

Reviewer 2:

The manuscript examines the ferroelectricity of water molecules confined to nano-channels of beryl through experiments. To avoid formation of hydrogen bonds between water molecules, the authors had experimentally placed water molecules in the channels of a beryl crystal, and the incipient ferroelectricity is observed. The authors attributed the ferroelectricity of confined water molecules to the disruption of hydrogen bond network and the intermolecular dipole-dipole interactions.

This work is of great interest and important.

Our reply.

We thank the Reviewer for his high appreciation of our results and positive evaluation of our manuscript.

Reviewer 2:

I support publication of this work in Nature Communications after the authors address the following comments:

Experimental evidence of placing water molecules in the structural channels of a beryl crystal must be provided in the supporting information.

Our reply.

As is written in the Methods section of the manuscript, this evidence is provided by the chemical analysis of the studied crystal performed before spectroscopic experiments. We have added to the section, that the presence of water molecules in the channels is evidenced by the known experimental material on the beryl crystals where relatively large water molecule can find enough space only within the channels' cages. It is also clearly demonstrated in our experiments by the observation of intramolecular ν_1 , ν_2 and ν_3 modes and of their $\nu_1 + \nu_2$ overtone; the values of the frequencies are only slightly shifted relative to those of a free H₂O molecule, evidencing its weak coupling to the crystal lattice that can be realized only when the molecule is within the cage.

Reviewer 2:

Discussion on how to put water molecules into the nano-channels of beryl since the bottleneck is very narrow (~2.8 Å).

Our reply.

This was specified on page 3 of the manuscript as:” Crystals grown in an aqueous environment contain water trapped in the framework of the crystal lattice in such a way that single H₂O molecules reside within the cages”. To make it more clear, we have completed the sentence with

“the molecules are captured into the cages during the growth process, i.e., during the cages formation”.

Reviewer 2:

Related experimental study could be cited (e.g. PNAS 108, 3481 (2011)).

Our reply.

We thank the Referee for pointing out that we missed important recent literature. We have added the citation to this work (Ref. [20]).

Reviewer 2:

The orientation profile of water molecules confined to nano-channels should be provided.

Our reply.

In our manuscript we cite a large body of literature where ample experimental results are given about the well-known orientation of the water molecules in beryl. The orientational profiles of the water molecules in our samples are evidenced also by our polarization dependent spectroscopic data. We now state that in the text as:

“The two orientational profiles of the molecules of both types in our crystals are unambiguously verified by our observations of H₂O intramolecular vibrational modes ν_1 , ν_2 and ν_3 that couple to the probing radiation strictly differently for the two orientations of the molecule, see Figure 3 in [28]”.

Replies to remarks of the Third Reviewer

Reviewer 3:

The authors in this work report incipient ferroelectricity of water molecules confined to nano-channels of beryl crystals, in which the intermolecular hydrogen coupling is strongly weakened or completely absent. Based on the theoretical and experimental investigation on the property of the water, the authors reveal that it is the intermolecular electrical dipole-dipole interactions, rather than by the hydrogen bonds as in bulk water or solid ice, plays a key contribution to the incipient ferroelectricity of the confined water. Hence, the work provides new insight in the ferroelectricity of the water. Owing to the ferroelectricity of confined water playing a significant role in various phenomena and areas of natural sciences, I'd like to recommend publication of the work in nature communications after major revision.

1) The authors claim in the work that "there is no firm experimental evidence so far for ordering within the subsystem of water molecules". In fact, there are some examples experimentally demonstrated existence of ferroelectric and antiferroelectric water in confined system. Because both are related to the order of water molecules, thus the claim should be revised and the related references should be cited.

Our reply.

First of all, we also want to acknowledge the positive assessment of our work by the Reviewer. We agree with the first comment of the Reviewer and have described the actual situation in some more details in our revised manuscript. We have added the following:

“As to the experimental realization of the dipolar ordering, the situation is not so unambiguous and clear. On the one hand, there seem to be firm indications towards ordered (ferroelectric or antiferroelectric) arrangements of the water molecules (ice nanotubes) within one-dimensional channels of carbon nanotubes or molecular organic structures [20 21] or on two-dimensional surfaces (ice slabs) [9 22 23]. On the other hand, either the fraction of the polarized dipoles can be very low, of the order of 1% or even smaller [22], limited to only few surface layers, or the reliability of the obtained results is discussed [8 19]”.

Reviewer 3:

2) How about the crystal size? I'd like to suggest the authors provide the picture of the crystal.

Our reply.

We thank the Reviewer for these useful suggestions. Accordingly we added some words on the crystal size to the Methods section; Fig.1 was amended by a photograph of a Beryl single crystal.

Reviewer 3:

3) From a statistical point of view, there is no doubt that each cage in the compound contains 0.3 water molecules. The actual situation may be that no water exists in part of the cage, since the water molecules can form doublets, triplets, etc., disposed in adjacent cages separated by the bottlenecks inside the channels as mentioned by the authors. Considered that the diameter of the cage is about 5.1 Å, is it possible to encapsulate one water molecule into the cage. If possible, I'd like to suggest the author to prepare the compound and investigate the property of the confined water.

Our reply.

We appreciate the insight of the Reviewer and his/her comments and suggestions, as they exactly describe what we have done in our experiments. We have grown a crystal of beryl that contained single water molecules in some cages with the filling factor of about 30%. These molecules are isolated, or grouped in dimers, trimers, etc., according to our statistical analysis (Methods section). Some cages do not contain water molecules at all; note however, that the cavity space is

not sufficient to host two molecules at the same time. This is the system we have actually studied.

We hope that the above explanation did answer the pertinent questions of the Reviewers. We are grateful for their comments and suggestions which we have incorporated into the corrected manuscript. We are confident that the present manuscript should be published in Nature Communications.

On behalf of the authors
with the best regards,

Martin Dressel

Reviewer 1:

I would like to thank authors for response to my comments, and I accept some of them. Nevertheless, I still have a question regarding the observed soft modes at about 20 and 40 cm^{-1} (2.5-5 meV) at 5 K, which increase their values to $\sim 60 \text{ cm}^{-1}$ (7.5 meV) at 300 K. In addition at low temperatures the authors observed sharp peaks in the range 1.2-1.5 THz (4.8-6 meV) which were explained as twist-like modes of the H_2O molecules. Recently, a paper on inelastic neutron scattering (INS) study of water in beryl has been published (Kolesnikov et al., Phys. Rev. Lett. 116 (2016) 167802), which presented observation of multiple tunneling peaks in the INS spectra in the energy range 2-118 cm^{-1} (0.27-14.7 meV) at low temperatures, which strongly decrease their intensity with temperature increase, but without noticeable change in the peaks positions. Unfortunately, the peaks observed in that paper are not visible in the manuscript under review, and the peaks in the manuscript at 20-40 cm^{-1} are completely missed in the INS spectra. Another related comment is regarding the estimated depth ($A=1.41 \text{ meV}$) for the water molecule rotational potential well. In the above mentioned paper on INS study of water in beryl the observed maximum splitting of the ground state is 14.7 meV, that means that the depth of the potential well should be larger than that value, so the value $A=1.41 \text{ meV}$ in the manuscript is at least one magnitude order smaller, than obtained from INS study. It is well known that the INS spectra are very sensitive to the vibrational modes of hydrogen due to anomalously large neutron scattering cross-section on hydrogen, and INS spectra directly related to the density of vibrational states. Based on these, it should be clarified or discussed in the manuscript why the observed soft modes (as well as twist-like modes) are completely missed in the INS spectra of water in beryl, and why there is a big discrepancy on the value of depth of the potential well. Could it be that the peaks observed in the manuscript are not related to the vibrational modes of water in beryl? Maybe water influences the beryl cage vibrations, which were observed in the manuscript? Therefore, I need to have answers for these questions, before I can make the recommendation for the publication.

Our replies.

1. The relatively intense mode around 90 cm^{-1} (11 meV) is clearly detected in both our THz and INS spectra, and its origin is firmly identified by us and by Kolesnikov et al., as being due to translational motion of the H_2O molecule along the c -axis.
2. There may be several reasons why the other absorption lines in the INS spectra (0.27, 0.66, 1.49, 1.63, 8.4, 12.7, 14.7 meV) are not identified in our THz-IR spectra:
 - a) There is no simple relationship between the strengths of polar modes in the dielectric spectra, where the intensities are determined by their dipole moments, and those in the spectra of INS. For example, some vibrations detected by the INS may not produce changes in the dipole moment of H_2O molecules within the cages strong enough to be detected by infrared spectroscopy. As for the infrared spectroscopy, the selection rule $\Delta m = \pm 1$ applies to the optical activity of the transitions between the tunnel-split energy levels [Gorshunov et al. J. Phys. Chem. Lett. **4**, 2015 (2013); Zhukova et al. J. Chem. Phys., **140**, 224317 (2014)]. By contrast, the INS spectroscopy is not restricted by these rules and it can observe more transitions, as specified in Table SI in the Supplementary material of Kolesnikov's paper.
 - b) The neutron scattering intensity is roughly proportional to the density of vibrational states, i.e. it integrates the vibrational response over all wave-vectors. Conversely, the optical (THz, IR) spectroscopies probe near-zero wave-vector excitations (close to the Γ -point of the Brillouin zone in case of crystals). Thus, some resonances seen in the INS spectra may not be observed by the THz-IR spectroscopy.
 - c) The intensities of most INS modes absent from our THz-IR spectra are considerably lower than that of the translational mode at around 11 meV clearly observed in both THz-IR and INS data. Some of these weak modes with higher frequencies (12.7 and 14.7 meV) could be

- “hidden” within the set of resonances in our spectra above 100 cm^{-1} (above 12 meV) [Gorshunov et al. J. Phys. Chem. Lett. **4**, 2015 (2013)].
3. As stated in our manuscript, the modes we observe in the range 1.2-1.5 THz are related to the librations of heavy oxygen. Since the INS spectroscopy is “sensitive primarily to the motion of individual hydrogen atoms”, as is rightly written in the paper of Kolesnikov et al. and as is well-known, it is not surprising that these terahertz modes are not observed in the INS spectra.
 4. The dynamics of the terahertz soft excitation we observe for the perpendicular geometry (the peak that softens from 60 cm^{-1} at 300 K down to 20 cm^{-1} at 2 K, Fig.5a in the manuscript) correspond quite well to the results of the INS measurements, namely to the broad wing below 5 meV (Fig.1a in the paper of Kolesnikov et al.). In the physics of ferroelectrics such a wing centered around zero frequency in the inelastic scattering spectra is called a central peak and the temperature evolution of its half-width closely traces the behavior of overdamped soft excitations that typically accompany the structural phase transitions, see e.g. A. D. Bruce and R. A. Cowley, *Structural Phase Transitions*, London: Taylor & Francis (1981). Though the INS data are influenced by averaging over all wave-vectors, the temperature evolution of the INS spectra of Fig.1a below 5 meV resembles quite closely the response of our soft mode. At 40 K, the THz soft mode is located near 20 cm^{-1} (2.5 meV) and its width (damping) is about the same (Fig.3b of our manuscript). At $T = 2\text{ K}$, the mode shifts down to $\sim 12.5\text{ cm}^{-1}$ (1.55 meV), its damping staying at about 20 cm^{-1} (2.5 meV), so the mode becomes overdamped. A basically identical response is seen in the INS spectra in Fig.1a: at $T=45\text{ K}$ there is a bump at $>2\text{ meV}$ of a width of 2-3 meV that gets narrower and more intense while cooling, similar to what we observe in the THz spectra. Unfortunately, Kolesnikov et al. did not analyze this interesting low-energy behavior in their paper.
 5. The final issue is the discrepancy in the potential well depth. Our value of $(1.41 \pm 0.05)\text{ meV}$ was obtained based on the model of Nakajima and Naya which we have modified to the case of a six-well potential and applied to our experimental spectra. A similar value of 0.83 meV was obtained independently by our *ab initio* DFT analysis carried out with the VASP packages with HSE03 hybrid functional. Both these values are also in a good agreement with the characteristic energy of the process that inhibits the ferroelectric phase transition, as determined from the fit using Barrett’s formula (Figs. 2, 7 and Eqs. 10, 11 of the manuscript), yielding $k_{\text{B}}T_1=1.7\text{ meV}$ (where k_{B} is the Boltzmann constant). This indicates that the dipole moments cannot align themselves macroscopically within the channels since they fall into different potential minima separated by barriers of about 1.7 meV. According to the general statistical principles applied to the system with a multi-minimum potential [Y. Onodera, Progr. Theor. Phys. **44**, 1477 (1970)], the Barrett-like leveling off of the soft-mode frequency during the order-disorder or displacive phase transitions happens when the ratio of the Curie temperature to the potential well depth gets close to unity (Fig.11 in [Y. Onodera, Progr. Theor. Phys. **44**, 1477 (1970)]). This again points to the value of 1.7 meV, based on the Curie temperature $|T_{\text{C}}|=20\text{ K}$ which we obtained experimentally from the temperature variation of the dielectric permittivity. Another confirmation of the validity of our DFT estimate of the potential well depth is provided by the closeness of the obtained value 1.33 meV of the dipole-dipole coupling energy to the energy $k_{\text{B}}T_{\text{C}}=1.7\text{ meV}$ characterizing the strength of dipole coupling. Finally, our DFT analysis has provided the frequencies and relative intensities of the infrared translational and librational modes of the trapped water molecules and of the isotopic shifts (when H_2O is replaced by D_2O); these values agree well quantitatively with our experimental data, as written in the manuscript. In contrast, the energy barriers of 48 meV and 56 meV stated by Kolesnikov et al. seem to be rather large and might imply quite a strong coupling of the H_2O molecules to the cages’ walls. Such a coupling could, in turn, lead to a blue shift of the intramolecular modes ν_1 , ν_2 and ν_3 relative to their positions in the free molecule. It is well-known, however, that the intramolecular vibrations frequencies of H_2O within beryl are very close to those of free molecules; see, for example, [L. M. Anovitz, E. Mamontov, P. ben Ishai, and A. I. Kolesnikov, Phys. Rev. E **88**, 052306

(2013); A. I. Kolesnikov, L. M. Anovitz, E. Mamontov, A. Podlesnyak, and G. Ehlers, J. Phys. Chem. B **118**, 13414 (2014)]. We have to conclude that the origin of the discrepancy remains unclear between the two quite different estimated values of the potential well depths.

Changes made

1. Page 6: we inserted
“Their resonance frequencies shift when H₂O molecules are replaced with D₂O [32].”
2. Page 12: we inserted
“The long-range ordering of dipoles along the channel can also be suppressed by the fact that during cooling down the dipoles occupy different potential minima of the six-well potential and cannot align due to the barriers of height $k_B T_1 = 1.7$ meV.”
3. At the end, before concluding part: we added a few lines discussing the recent Phys. Rev. Lett. by Kolesnikov et al.
4. We added discussions with A.Zhugayevich and J.Hlinka to the Acknowledgements.

Our replies to the comments of the Reviewer.

Reviewer #1 (Remarks to the Author):

I thank the authors for the answers and the manuscript modification, but I still disagree with some of their statements.

Regarding the observed resonances at 1.2-1.5 THz, which "are connected with the twist-like modes of the H₂O molecules that involve librations of the oxygen ions [32]." Twist-like mode of water molecule is a libration of water molecule around its dipole moment (see e.g. https://en.wikipedia.org/wiki/Molecular_vibration), therefore this mode should be inactive in optical spectroscopy since it does not affect the dipole moment of the molecule. Also, the twist mode does not involve vibration of oxygen at all. These resonances could be, hypothetically, due to a wagging mode (librations of water-I molecule around the axis of channels / c-axis of beryl).

Our reply:

The Reviewer is right about the labelling of the molecular vibration we observe. The vibration is more like a wagging type. We have corrected corresponding part of the text, and thank the Referee for pointing out our mistake.

Reviewer #1

In this case the 1.5 THz (6 meV) excitation would require the potential depth (A) to be larger than this value, which disagrees with the estimates made in the manuscript, A=1.41 meV. In addition, eigenvectors of water librational modes are much larger for hydrogen than for oxygen, therefore INS spectra for water librational modes are mostly originated due to neutron scattering on hydrogen, and these modes are the most strong in the water INS spectra.

Our reply:

The connection between the quantum energy levels of the water molecule in beryl (and, correspondingly, the frequencies of the transitions between them) and the depth of the potential well experienced by the molecule are described in details in our previous publication [Zhukova et al. J. Chem. Phys., **140**, 224317 (2014)] and in the subsequent work of Kolesnikov et al. [Phys. Rev. Lett. **116**, 167802 (2016)]. The key point is that the water molecule in beryl cannot be regarded as a pure harmonic oscillator. Due to tunneling, the molecular states types are intermediate between (or a mixture of) the harmonic oscillator states within the wells quantized as

$$E_n = \left(n + \frac{1}{2}\right) \hbar\omega_0, \omega_0 = 6\sqrt{\frac{A}{2I}}, n = 0, 1, \dots, \text{ and the states of a free rotator with}$$

$E_m = Bm^2, B = \frac{\hbar^2}{2I}, m = 0, 1, \dots$. Within this approach, for the parameter A=1.41 meV that was determined in our experiments one obtains the tunnel splitting of around 6 meV, in perfect correspondence with the positions of the observed terahertz resonances.